# A scoping review of community-based mental health intervention for children and adolescents in South Asia

Rhiannon A. Willmot[1] , Rebecca A. Sharp[1] , Azlina Amir Kassim[1] and John A. Parkinson[2]

[1]School of Human and Behavioural Sciences, Bangor University, Bangor, UK and [2]College of Human Sciences, Bangor University, Bangor, UK

## Overview Review

mental health; South Asia; LAMIC; children and adolescents; community-based intervention

**Corresponding author:**
Azlina Amir Kassim,
E-mail: a.amirkassim@bangor.ac.uk

### Abstract

Children and adolescents in South Asia are exposed to significant mental health risks. Yet, policy to prevent or treat youth mental health problems in this context is underdeveloped, and services are difficult to access. Community-based mental health treatment may offer a potential solution, by increasing resource capacity in deprived settings. However, little is known about the current community-based mental health provision for South Asian youth. A scoping review was conducted across six scientific databases and hand searching of reference lists to identify relevant studies. Study selection and data extraction were performed by three independent reviewers using predefined criteria, an adapted version of the template for intervention description and replication checklist and the Cochrane Risk of Bias Tool. The search identified 19 relevant studies published from January 2000 to March 2020. Studies most frequently addressed PTSD and autism, were conducted in India and Sri Lanka, used education-based intervention and were based in urban school settings. Community-based mental health provision for the South Asian youth is in its infancy, but holds promise for providing essential resources to treat or prevent mental health disorder. New insights on approaches are discussed, which are valuable for South Asian settings, primarily task-shifting and stigma reduction, with implications for policy, practice and research.

### Impact statement

Life for children and adolescents in South Asia can be challenging, with many reported risks to mental health. For example, various physical (natural disasters and forced displacement), social (crime and neglect), contextual (gender disparity and child labour) and environmental (substance exposure, poor access to services) factors can compromise the well-being of young people. There is also a lack of available services to support children in the face of adversity. Our scoping review intends to improve this situation by investigating what is currently being done to help children and young people in South Asia live well. We provide an overview of mental health risk factors, and focus on community-based intervention as a means of reaching those who are geographically or financially distanced from formal services. We present stories of success, and implications for future work, with task-shifting and stigma reduction methods identified as particularly valuable. These findings provide important information not only for those attempting to boost youth mental health in South Asia, but also for other low-resource settings. Here, approaches that mobilise the local population to address poor mental health and use available resources well can be most effective. We also present information on factors that dictate the feasibility and acceptability of interventions amongst South Asian communities. Our recommendations for practice provide specific detail on how interventions can achieve the greatest likelihood of success, including the use of targeted specialist intervention and comprehensive supervision from professionals. Our recommendations for research cover the increasing development of research capacity in rural locations, the use of strength-based approaches in reducing stigma and additional factors to enhance the feasibility of task-shifting. Collectively, these insights can be applied at the global level, to enhance both research and intervention focusing on the mental health of children and young people.

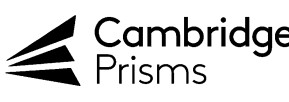



### Introduction

The need to promote youth mental health in South Asian countries is significant and pressing. Globally, there is a lack of sufficient care to meet the mental health needs of children and adolescents (Kieling et al., 2011), a disparity which is particularly distinct in many of the low- and middle-income countries (LAMICs) in South Asia. Although in some South Asian regions a

distinction is drawn between psychiatric disorders (e.g., schizophrenia), externalising disorders (e.g., attention-deficit hyperactivity disorder), internalising disorders (e.g., depression) and neurodevelopmental disorders (e.g., autism), in other locations, all conditions are bracketed under the umbrella term of 'mental disorders'. The rate of general mental disorders among South Asian children and adolescents ranges from 1.81 to 23.3% (Hossain et al., 2020). Little research has addressed the prevalence of specific disorders amongst young South Asians; however, one study indicates between 3.9 and 25.4% of adolescents experience suicidal ideation (Aggarwal and Berk, 2015). Autism spectrum disorders (ASDs) affect up to 1 in 93 children (1.1%) across Bangladesh, India and Sri Lanka (Hossain et al., 2020). The prevalence of ASD amongst children and adolescents in Pakistan, Nepal and Afghanistan is unknown. Substantial variations in prevalence figures indicate that further work is necessary to establish precise estimates for specific disorders and populations.

Whilst mental health infrastructure in these regions is generally lacking (Hanlon et al., 2014; Angdembe et al., 2017), services for young people are particularly underdeveloped. In Bangladesh, Sri Lanka, Nepal and Pakistan, there are no child-specific mental health policies (WHO, 2018; Hossain et al., 2021). Additionally, very few psychiatrists offer specialised treatment for children or adolescents in India, Bangladesh, Pakistan, Nepal or Afghanistan (Agarwal, 2021; Hamdani et al., 2021; Skuse, 2021; Singh et al., 2022; Wang et al., 2022). Indeed, many countries in South Asia follow the 'medical model' of mental health; support is predominantly provided by hospital-based psychiatrists, with very few outpatient or community-based facilities, particularly in Nepal, Pakistan, Afghanistan and Bangladesh. However, these services are more widely available in Sri Lanka and India. Relatedly, with the increasing trend of immigration, relatively poorer South Asian regions suffer from 'brain drain', where skilled professionals often migrate to more developed nations (Mullan, 2005), resulting in a lack of psychiatrists, clinical psychologists and social workers (Thara and Padmavati, 2013). This phenomenon also leads to a gap in supervising and supporting early career researchers who often need guidance to develop their skills (Sharma and Razzaque, 2017). The lack of mental health specialists for supervision has been identified as a key concern towards capacity building (Eaton et al., 2011).

Targeting the well-being of young people in South Asia is pertinent, because such a strategy offers benefit beyond childhood and adolescence. A substantial proportion of mental health disorders originates in early life (Kim-Cohen et al., 2003; Kessler et al., 2007), and interventions to reverse or prevent the effects of cognitive, social and economic deficits reap stronger benefits when delivered to the young (Institute of Medicine and National Research Council, 2000). As such, methods that support the development of lifelong psychosocial resources are a cost-effective route to service provision. Moreover, youth mental health is particularly important in low-resource South Asian regions, because proactively fostering the collective well-being of deprived societies helps to avoid the entrenchment of development problems, and augments global prosperity (Kieling et al., 2011).

There is a growing evidence base regarding the epidemiology of mental health disorder in children and adolescents across South Asia. Established risk factors for the development of emotional, intellectual and behavioural disorders in LAMIC South Asian settings include low educational stimulation (Arun and Chavan, 2009), gender disparity (Rudatsikira et al., 2007), being orphaned or raised in an institution (Ruiz-Casares et al., 2009; Erol et al., 2010),

abuse or neglect (Benjet, 2010) and exposure to trauma including harmful substances (Roy et al., 2009), violence (Panter-Brick et al., 2009), armed conflict and war (Qouta et al., 2008; Harel-Fisch et al., 2010; Layne et al., 2010; Oldham-Cooper et al., 2011), forced displacement (Mels et al., 2010), immigrant status (Wong et al., 2010) and natural disasters (Jia et al., 2010; Li et al., 2010). The most pertinent risk factors during infancy and early childhood are limited care and home stimulation (Grantham-McGregor et al., 2007), exposure to violence (Walker et al., 2007) and poor maternal mental health (Anselmi et al., 2004). During the school-age period (from ages 5 to 18 years), factors of low physical health (Lo et al., 2009), academic difficulties (Arun and Chavan, 2009), bullying (Chaux et al., 2009), family dysfunction (Lee et al., 2011), child labour (Ibrahim et al., 2019), physical, sexual and substance abuse (Benjet, 2010; Curto et al., 2011; Miller et al., 2011), pathological use of the Internet (Lam and Peng, 2010) and teenage pregnancy (Miller et al., 2011) can also give rise to problematic mental health. These factors and their relationships to mental health risk are observed in numerous settings across the globe, and the aforementioned articles evidence their specific relevance to the development of mental health conditions in South Asia. An overview of the range of contributory factors is conceptualised in Figure 1.

Whilst many mental health risks are highly prevalent in South Asia, particular psychosocial and environmental conditions can mitigate against the development-related disorders. Evidence suggests that in LAMIC South Asian settings, factors that contribute to this kind of resilience are effective behavioural and emotional self-regulation (Goldstein and Brooks, 2013), emotionally responsive and competent parenting, educational resources, strong carer attachment and a comprehensive peer network (Wyman et al., 1999; Kieling et al., 2011).

The development of mental health services to address and prevent identified mental health disorders in South Asia aligns with the global health agenda (Department of Health and Social Care, 2019). However, the progress of this goal is complicated by a number of cultural and contextual factors. For example, a comparatively low research output in South Asian countries hinders the development of policy and practice (Tomlinson et al., 2014), which in turn complicates the identification of effective resource deployment. Despite a predominantly rural population, the majority of mental health services are located in large cities, which exacerbates issues related to a general lack of specialist personnel and facilities (Palit and Bandyopadhyay, 2016). This is particularly pertinent given specific observations that diagnostic and therapeutic services in primary care are lacking in India and Pakistan (Agarwal, 2021; Hamdani et al., 2021), and that in Nepal, primary care practitioners are prohibited from legally diagnosing mental health conditions (Angdembe et al., 2017). Collectively, this means individuals are unable to obtain appropriate treatment from more localised forms of healthcare.

Further complication arises from the use of healthcare systems predominantly based on out-of-pocket payments in Bangladesh, India and Nepal, which exclude a majority of residents from obtaining formal treatment (Maselko et al., 2016; Angdembe et al., 2017). Rates of urban poverty and unemployment are significant and increasing across LAMIC South Asian countries, therefore not only giving rise to mental health problems, but also restricting the capacity of those affected to receive treatment (Patel et al., 2008). In Pakistan, some support is offered by specialist taxes known as 'zakat', which is used to provide national social assistance to those living in poverty. However, the collection and distribution of zakat is disorganised, rendering it inaccessible to most of the

**Figure 1.** Contributory factors in the development of mental health problems amongst children and adolescents in South Asia.

population (Karim et al., 2004), and residents are expected to pay at least 20% towards the cost of expensive services and medicine. Rates of mental health disorder tend to be the highest in the most deprived groups, as a result of food scarcity leading to malnutrition, and co-existing factors of family dysfunction and violence, criminality and neighbourhood danger (Patel et al., 2008). As such, those who are most in need of mental health treatment are also those least able to access it.

Lastly, the tendency to stigmatise and discriminate against people with mental illnesses is widespread across Asia (Zhang et al., 2019). There is little awareness and education on the complexities of mental health problems, which are known moderators of stigma (Corrigan et al., 2014). People with mental health problems are often perceived as dangerous or violent, and some evidence suggests that even the attitudes of professionals working to improve mental health are discriminatory (Lauber and Rössler, 2007). Stigma can also present a barrier to the scaling-up of successful mental health services (Angdembe et al., 2017), and directly affects not only people living with mental illnesses, but also their wider support system and willingness to seek treatment (Corrigan et al., 2014). Collectively, these factors impede the delivery of care, which is already complicated by issues of limited resource, low accessibility and economic cost. Relatedly, Roberts et al. (2020) report that in rural areas of India, there is low perceived need for treatment and low awareness regarding psychological disorders. For example, depressive symptoms were described as 'tension', which are a natural response to living in poverty, or having a physical illness, and distinct from mental illness. This leads to the belief that such will only be alleviated if a

person's socioeconomic situation improves or symptoms of their physical illness are treated. As a result, there was a low take-up for treatment of depressive symptomatology, even when services were expanded.

In answer to the difficulties of facilitating effective mental health treatment in South Asia, the World Health Organization (WHO) has endorsed the value of community-based programmes (WHO, 2011). Such programmes include all those that take place locally, for example, in schools, primary care and community centres. Some community-based programmes may also help facilitate nonspecialist personnel to deliver mental health provision in their own environment, by offering training, supervision and support. This provides an immediate solution to the lack of available mental health specialists, and can drastically promote the availability of essential treatment to distanced, impoverished and marginalised populations (Angdembe et al., 2017). This practice is one example of 'task-shifting', and can increase the reach of programmes to a broad range of locations, whilst also making mental health support more financially accessible for both individuals and whole societies (WHO, 2007). The local nature of community-based programmes, as well as harnessing of pre-existing resources, can help target individuals before they begin to exhibit stark mental disorders, which require more extensive and expensive treatment to reduce the overall cost of delivering mental health services (Zraly et al., 2011). This style of care can thus precede the necessitation of psychotropic drugs, which bypasses the complication of specialist diagnosis and prescription (Angdembe et al., 2017).

Community-based services have been shown to reduce the impact of stigma. For example, work in Korea has demonstrated

the role of community-based services in promoting mental health literacy amongst a broader proportion of the population, and in turn increasing the degree of respect devoted towards people with mental illnesses (Zhang et al., 2019). Additionally, community-based services use medical diagnostic terms to a lesser extent than formal healthcare services, meaning they are less likely to evoke prejudiced perceptions of service users (Corrigan et al., 2014). Community-based services are more easily accessible, meaning those with mental health problems may be able to receive treatment at an earlier stage when their difficulties are less pronounced (Lauber and Rössler, 2007).

Whilst community-based interventions present a practical and effective strategy to deliver evidence-based mental health treatment in low resource settings, little is known about their nature and implementation across South Asia for children and adolescents. Developing further knowledge on this topic can enable collaboration and integration by health professionals and policy makers to prioritise relevant research activities and services for LAMIC regions provide information on the fidelity and effectiveness of care in community-based settings and enable the potential scale-up of successful programmes.

Given the current lack of empirical evidence on the provision of community-based mental health programmes for young people in South Asian countries, and the complexity of this topic, we have chosen to present a scoping review with clear implications for research and practice, according to Daudt et al. (2013). This will provide a foundational summary of existing research, with a view to catalysing further investigation that considers more intricate themes across and within geographical and cultural contexts. As such, our primary objective was to identify the extent and type of evidence regarding mental health programmes for children and adolescents in South Asia, as well as identify gaps in knowledge. Secondary aims were to acknowledge examples of successful programmes with potential for delivery at scale, recognise principles to incorporate amongst new models of treatment and develop recommendations for integration across community-based South Asian healthcare.

## Method

### Positionality statement

Although this review is based in South Asia, it was conducted solely in the United Kingdom. Therefore, it would be pertinent to understand our personal and professional positionalities which have shaped this review. The first, second and last authors of this scoping review are White British academics from the United Kingdom. The third author is an ethnic South Asian academic from Southeast Asia with lived experiences in Southeast Asia and strong ties to South Asia.

All authors were a part of the Mental Health Initiative for South Asia (MhiSA), a 3-year project formed in 2018 with the objective of promoting science and research partnerships between the countries of South Asia and the United Kingdom. Throughout 3 years, MhiSA received funding from the British Council and support from other institutions such as the Ferdowsi University of Mashhad to conduct workshops and events that brought delegates from South Asia and the United Kingdom together and provided a platform to enable discussions and facilitate collaborations. One such workshop was the 'United Kingdom–South Asia collaboration on Mental Health Workshop' held in Nepal in April 2018. This workshop brought together policy makers, researchers and clinicians in the South Asia region and researchers in the United Kingdom to develop an agenda and strategy for mental health research and practice. Throughout activities and discussions during the events, one key theme that was identified was the need for more research in child mental health. We recognize that the aim of this project is to improve understanding of community-based mental health treatment in South Asia, and we have ensured that the study is unbiased, sensitive and appropriate to the South Asian context.

### Data selection

Between August 2019 and March 2020, the following databases were used to conduct an electronic search of relevant articles: Applied Social Sciences Index and Abstracts, Cumulative Index of Nursing and Allied Health Literature, Global Health, PsychInfo, PubMed and Web of Science. Articles were returned if they contained appropriate descriptors of mental health (e.g., 'depress*' and 'resilien$'), treatment (e.g., 'interven$' and 'therap$'), setting (e.g., 'community?base*' and 'non?government& organi?ation'), population (e.g., 'adolescen$' and 'school?age') and location (e.g., 'Pakistan$' and 'Sri Lanka*'). An example of the search strategy applied to the databases is provided in the supplementary materials. Hand searching of reference lists pertaining to included articles was also conducted. Given the funding source, we addressed research specifically conducted in Bangladesh, India, Afghanistan, Pakistan, Iran, Sri Lanka and Nepal, according to the British Council's definition of South Asia. Articles were restricted to those published between 1 January 2000 and the date of search.

### Inclusion and exclusion criteria

The title and abstract of each article were used to determine initial relevance. Of interest were articles which could inform future research on community-based psychosocial treatment of child and adolescent mental health problems across South Asia. Accordingly, all research designs were retained. However, research related to diagnostic protocol or prevalence estimates, and research regarding purely pharmacological interventions, were excluded. Studies conducted in non-community-based settings were also excluded, as were articles which only addressed adults over the age of 18 and those which concerned any population residing outside of South Asia, as defined in this review. If the study regarded a mixed-age sample, inclusion for review necessitated a sample mean age within 0 to <18 years. All studies were required to include mental health as a primary consideration. Articles describing treatments which were only indirectly related to mental health (e.g., substance abuse) were excluded. However, no limitations were placed on dose, duration or mode of delivery of mental health treatment. Book chapters and editorial articles were also excluded.

### Data analysis

Data for each included article were extracted using an adapted version of the template for intervention description and replication (TIDieR) checklist (Hoffman et al., 2014), leading to the identification of the following: number and age of participants, location of research, aim, type, duration and setting of intervention, sampling strategy, study design, comparators, outcome measurement, data analysis, results, conclusions, implications

and economic considerations. Where possible, articles were coded for risk of bias according to the Cochrane Risk of Bias Tool for randomized controlled trials (RCTs) and non-RCTs, and any additional limitations of each study were identified. Moreover, factors that supported or inhibited the success of intervention design, delivery and implementation were noted.

### Intercoder agreement

A second coder screened randomly selected articles ($n = 295$; 40%) to provide coder agreement data on inclusion and exclusion for review. Agreement was calculated by dividing the number of agreements (i.e., both coders elected to include or both elected to exclude the same article) by the total number of articles reviewed by both researchers ($n = 295$). Coders agreed upon 97% of articles, and for those upon which researchers did not agree, a discussion and mutual decision on inclusion took place. One coder then extracted data on all 26 variables the 19 articles, and one further coder conducted data extraction on all 26 variables for 10 of the 19 articles (53%). In this instance, agreement was defined as the extraction of exactly the same information from an article, and calculated via dividing the total number of agreements across variables and articles by the total number of variables ($n = 260$; 26 variables for each of the 10 articles). Agreement for these data was 96% (range 88–100% across variables).

### Results

The final literature search resulted in 19 studies which were included (Figure 2). An overview of the 19 studies with respect to their design, location, setting, aims and intervention is provided in Table 1.

### Delivery methods and personnel

Twelve articles involved mental health professionals, eight recruited parents or teachers and eight included community volunteers. Three studies reported on the production of workbook materials, one recorded the use of training via video conferencing and four interventions involved game-related components.

### Aims

The most commonly occurring intervention aims were to treat PTSD ($n = 6$) and autism ($n = 5$). Additional studies attempted to address behavioural problems ($n = 3$), anxiety ($n = 3$), cognitive development ($n = 2$) and general emotional problems ($n = 2$). Other articles concentrated on stress ($n = 2$), depression ($n = 2$), ADHD ($n = 1$) and suicide prevention ($n = 1$). Other research adopted a strengths-building approach, focusing on the promotion of general well-being ($n = 2$), resilience ($n = 1$) and self-esteem ($n = 1$). Five articles employed an educational approach as

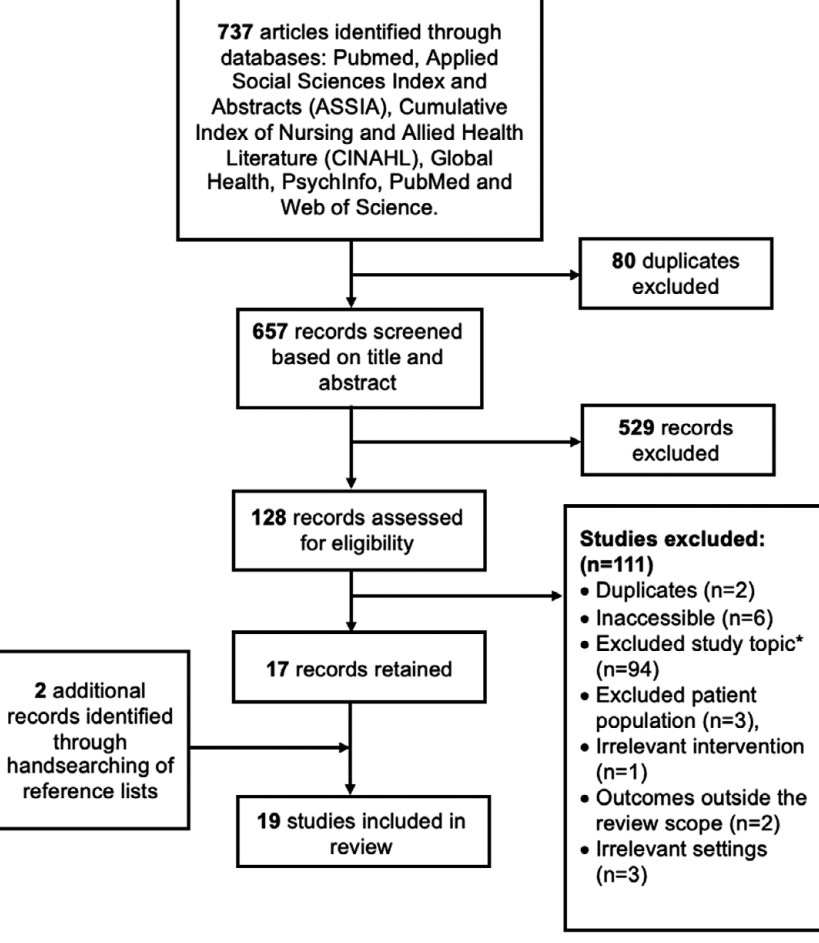

**Figure 2.** Flowchart depicting the selection of studies to be included in the review. *'Topic' refers to research related purely to diagnostic protocol or prevalence estimates rather than treatment.

**Table 1.** Overview of design, participants, setting, aims and intervention data across all identified studies ($n = 19$)

| Author(s) | Design | Population (n = sample size) | Setting | Treatment focus | Intervention | Comparator | Outcome measures | Results and conclusions |
|---|---|---|---|---|---|---|---|---|
| Abdollahian et al. (2013) | RCT | Children aged 7–9 years, diagnosed with ADHD $n = 30$ | Outpatient clinic Iran | ADHD | Sixteen-week programme of 45-min group-based game play designed to improve attention, memory teamwork, self-control, impulsivity and organisation, delivered bi-weekly. | No intervention | Pre-validated questionnaire | Play therapy effective in reducing symptoms of ADHD and expending energy which reduced hyperactivity. |
| Adhikari, Upadhaya, Satinsky, Burkey, Kohrt and Jordans (Adhikari et al., 2018) | CBA | Children aged 6–15 years, screened for behavioural problems screened using the Disruptive Behaviour International Scale-Nepal Version (DBIS-N) $n = 104$ (intervention) and 39 (follow-up). The DBIS-N measures locally defined child behavioural problems | School and home Nepal | Behavioural problems | Multilevel intervention to support parents and teachers in recognising and managing behavioural problems. Level 1: group discussion on assessing and responding to externalising behaviours delivered by counsellors to parents and teachers. Level 2: development of 4–6 person education and parent support groups by a psychosocial counsellor. Level 3: 1–3 home visits from a counsellor to support training, and progress of behaviour modification techniques (goal behaviours and reward system). | None | Pre-validated questionnaire and interview | Significant reduction in number and intensity of problems, and improved functioning. High acceptability from teachers and parents. |
| Andrew et al. (2020) | CBA | Children aged 10–20 months, living in urban slums $n = 421$ | Home India (urban slums) | Cognitive development | Eighteen-month programme of weekly 1-h home visits from a trained community worker to improve interactions between mothers and children using low-cost homemade toys. | No intervention | Adapted questionnaire and pre-validated questionnaire | Significant improvement in cognition, language and fine motor development. Improved quality of home environment. Low attrition, high take-up and high compliance. |
| Antle et al. (2018) | CBA | Girls aged 5–11 years, attending an NGO-funded school and with experience of trauma $n = 21$ | School Nepal (urban) | Anxiety, cognitive development and emotional problems | Delivery of a computer-tablet game designed to support attention regulation by school counsellors over 6 weeks. | Waitlist | Open-question survey and purpose-designed questionnaire | Significant improvement in self-regulation of calmness and attention, maintained to 2-month follow-up. High compliance. |
| Beckerf (2007) | Descriptive review | Child survivors of the 2004 tsunami in India $n = $ not listed | School, home and refugee camp India | PTSD and response to trauma | Delivery of counselling and workbook materials via community-level workers, | None | Not listed | Mental healthcare including psychological processing, control of |

(*Continued*)

| Author(s) | Design | Population (*n* = sample size) | Setting | Treatment focus | Intervention | Comparator | Outcome measures | Results and conclusions |
|---|---|---|---|---|---|---|---|---|
| | | | | | educated by NGO workers, teachers and local healthcare providers according to a train-the-trainer model. | | | stress, re-establishing of social connections and a sense of security recommended for disaster relief efforts. |
| Berger and Gelkopf (2009) | RCT | Survivors of the 2004 tsunami in Sri Lanka, aged 9–15 years *n* = 166 | School-based Sri Lanka (rural) | Anxiety, behavioural problems, cognitive development, depression, PTSD and stress | Twelve-week programme of 90-min sessions employing psychoeducational materials, CBT, meditation, art therapy, narrative techniques and homework to build resilience after trauma. Delivered by teachers who received 8 h of training, and supervised by local mental health professionals to ensure fidelity. | Waitlist | Pre-validated questionnaire and purpose designed questionnaire | Significant improvement on PTSD severity, functional problems, somatic complaints, depression and hope. Reduction in possible PTSD classification cases. |
| Blake et al. (2017) | CBA | Children in the lowest Bangladeshi socioeconomic tertile, aged 7–9 years and diagnosed with ASD *n* = 10 | Community centre Bangladesh (rural) | Autism | Development and delivery of educational materials and training to parents of children with autism over a 1-day group session and two 1:1 follow-up visits by a native clinician. | None | Informal interview and open-question survey | Parents reported appreciation of 1:1 training and group-based sessions, and intent to continue practicing skills learnt. High acceptability of programme across parents and clinician. |
| Borja Jr. et al. (2019) | Narrative analysis | Rohingya children aged 4–17 years who are refugees living in Bangladesh *n* = not listed | Refugee camp Bangladesh | Resilience and response to trauma | Needs assessment and delivery of psychological first aid via community-based support including recreational activities and structured play. Referrals to relevant services. Educational sessions for caregivers and teachers on child protection, health, nutrition and learning. | None | Not listed | Efforts to increase awareness, availability and acceptability of mental health services including relaxation spaces and techniques, structured play and counselling as important. |
| Chase and Bush (2002) | Case study | Two school children who have experienced psychological distress CS1: Male aged 11 CS2: Female aged 12 | Community centre Sri Lanka (urban) | Emotional problems and response to trauma | After school programme designed to promote ethnic reconciliation and recovery from trauma by providing child-focused activities including creative | None | Observation | Improved emotional, social and behavioural functioning. Improved acceptance of trauma and signs of resiliency. Increased sense of self |

| Author(s) | Design | Population (*n* = sample size) | Setting | Treatment focus | Intervention | Comparator | Outcome measures | Results and conclusions |
|---|---|---|---|---|---|---|---|---|
| | | | | | play and art. Staffed by local young adults trained by NGOs. | | | but some remaining psychological conflicts. |
| Divan et al. (2019) | RCT | Children aged 2–9 years, diagnosed with ASD using the INCLEN Diagnostic Tool for Autism Spectrum Disorder *n* = 40 | Home India (rural) | Autism | Lay health workers trained over 10 days by clinicians to conduct 1:1 home-based sessions with parents. Sessions are designed to increase parent–child communication via recognising and responding appropriately to child's interest and language levels. | Treatment as usual | Observation and pre-validated questionnaire | High feasibility reported by health workers. High acceptability from families. |
| Duggal et al. (2020) | Descriptive | Practitioners for children with developmental challenges *n* = 11 | Outpatient clinic India (online delivery) | Autism | Six-month programme designed to upskill professionals working with children who require developmental support. Personalised training delivered online, regarding the use of assessment tools and treatment plans. Structured around active learning, hands-on coaching, guided design, didactic teaching, observation, role play and group discussion. | None | Semistructured interview | Use of blended training format and combined online and face-to-face teaching appreciated and increased accessibility. Modular framework enabled flexibility around working commitments and direct implementation. High acceptability. |
| Gonsalves et al. (2019) | Descriptive | Children aged 12–17 years, who are identified as 'at risk' of selected mental health disorders *n* = 50 | School India (urban) | Anxiety, behavioural problems and depression | Development of a smartphone game designed to support mental health, delivered in tandem with low intensity in-person counselling. | None | Focus groups | Gamified and narrative elements appealing. Acceptable and feasible in sample. Consideration of population literacy, education and access recommended as important. |
| Jordans et al. (2011) | Descriptive | Children aged 7–15 years living in active or post-armed conflict settings *n* = 2,664 | School Sri Lanka (urban) | Response to trauma and stress | Multilayered system to assess, triage and treat children in a resource efficient manner. Secondary aims to promote awareness of mental health problems and reduce stigma across primary healthcare workers. | None | Purpose designed questionnaire | Reported reduction in behavioural and emotional problems and improvement to emotional and social functioning in children. Significant distress reported by service providers. |

| Author(s) | Design | Population (*n* = sample size) | Setting | Treatment focus | Intervention | Comparator | Outcome measures | Results and conclusions |
|---|---|---|---|---|---|---|---|---|
| Palit and Chatterjee (2006) | Descriptive | Parents of children aged 6–9 months with cerebral palsy and comorbid disabilities *n* = 50 | Outpatient clinic India (rural and urban) | Cerebral palsy with comorbid disabilities | Ninety-minute weekly sessions delivered as required by and to parents of children with cerebral palsy. Experienced parents provided with brief training, before delivering sessions to nonexperienced parents on understanding and interacting with their child, areas of prognosis, landmarks of success and emotional regulation. Designed to educate and promote hope via personal experience. | None | Purpose designed questionnaire | Improved mental health, confidence and ability to care for child across parents. Appreciation of parent-to-parent discussion, and reduced frustration, anxiety and helplessness. |
| Perera et al. (2016) | RCT | Children aged 18–40 months diagnosed with ASD using DSM-IV *n* = 104 | Home Sri Lanka (urban) | Autism | Two-month programme designed to promote joint attention across caregivers and children. Caregivers provided with 7 h of training using workbook materials and video clips, as well as support on using material available at home. Caregivers instructed to conduct 1:1 face-to-face play activity for 2 h per day delivered in 20–30-min blocks, and trained using workbook progress review sessions conducted monthly. | Different population | Observation | Significant improvement on parent–child interaction. High feasibility of training parents in the role of therapist. |
| Puthanveedu and Sekar (2019) | RCT | Children aged 10–16 years 'at risk' of low self-esteem *n* = 184 | Community centre India (rural) | Self-esteem | Four-year programme conducted according to a train-the-trainer model, whereby stakeholders train NGOs, who in turn train community members to deliver psychosocial care around seven key topics; life skills, enriching family life, physical health, study and learning skills, alcoholism, adolescent girls problems and mental health. | Different population | Pre-validated questionnaire | Significant improvement in self-esteem observed across all groups. Intervention demonstrated efficacy regardless of social status or gender. |

**Table 1.** (*Continued*)

| Author(s) | Design | Population (*n* = sample size) | Setting | Treatment focus | Intervention | Comparator | Outcome measures | Results and conclusions |
|---|---|---|---|---|---|---|---|---|
| Rahman et al. (2016) | RCT | Children aged 2–9 years diagnosed with ASD using INCLEN = 65 | Outpatient clinic and home India and Pakistan (urban) | Autism | Six-month programme designed to support parents in developing autism appropriate communication using feedback on videoed parent–child interactions. Delivered via fortnightly 1:1 clinic or home-based sessions, between a health worker and parent, with child present. Conducted according to a task-shifting model, whereby UK researchers train local specialists, who in turn train implementation therapists with no prior experience of mental healthcare. | Treatment as usual | Observation | Significant improvement on parental synchronous interaction and child communication initiations. Negative effect on shared attention. High adherence and low attrition. |
| Rajaraman et al. (2012) | Descriptive | Children aged 9–17 years recruited from nine schools *n* = 4,303 | School India (semi-urban) | General well-being | Two-year intervention delivered by lay school health counsellors to improve mental and physical health, using life skills training and individual counselling. Establishment of a 'speak out' box, to encourage participants to anonymously submit information on personal problems, for large group discussion. | None | Purpose designed questionnaire | Practical improvement to health environment. Instigated action on bullying, extreme disciplinary practices and sexual health. Adoption of anger management and emotional coping skills. Initial stigmatisation and resistance; however, high and increasing acceptability across students and teachers over 2 years. |
| Zachariah et al. (2018) | Descriptive | Children aged 12–15 years, acting as peer educators across six schools *n* = 76 | School India | General well-being and suicide prevention | Eight-month programme in which school children are trained and supported in acting as peer educators on suicide prevention. Training includes five modules: human capacity for response, self-care, active listening, understanding suicide, ways to respond, and supplemented with weekly 45-min group sessions to share experiences with other peer educators and adult NGO workers. | None | Semistructured interview | Improvement across attitudes and ability to support those in distress, as well as bonding across PEs. Variable ability to postpone judgement. Improved emotional development and independence. |

Abbreviations: ASD: autism spectrum disorder; CBA: Controlled before-and-after

a means of improving mental health awareness. Equally, 12 articles used educational methods to train a broad range of skills relevant for the management of mental health conditions that across articles, from self-regulation to communication, and was directed at both those at-risk or experiencing a mental health problem, as well as parents, caregivers and healthcare professionals responsible for providing support. Task-shifting methods were also employed in 16 articles, involving the training of teachers, parents, volunteers, lay health workers and counsellors, local specialists, parents and care-givers and children themselves. Three of these articles employed task-shifting via the specific adoption of a train-the-trainer approach. Finally, eight articles considered the development of context specific resources, including technologies, workbooks and games suitable for specific cultural and demographic requirements.

### Risk of bias and critical appraisal of sources of evidence

Of the studies able to be assessed using the Cochrane Risk of Bias Tool for RCTs and non-RCTs ($n = 6$), bias for sampling randomisation, allocation concealment, participant and outcome blinding and data completeness and reporting was generally low (see Table 2). In one study (Antle et al., 2018), a lack of allocation concealment and blinding of participants presented a high risk of bias, and four of the six studies lacked sufficient information to identify risk of bias on one or more of the assessment domains. The most frequent limitations of quality were a small or poorly representative sample size, lack of suitable comparators, low use of pre-validated questionnaires and little assessment of longitudinal outcomes. Additionally, most intervention programmes were designed and evaluated by the same researchers, with little use of empirical frameworks to increase the validity of findings. Alternatively, several studies presented robustly designed and well-validated research (Berger and Gelkopf, 2009; Rahman et al., 2016; Divan et al., 2019; Andrew et al., 2020), including one qualitative analysis which used multiple perspectives to substantiate information and triangulation (Zachariah et al., 2018). As such, the results of some studies, particularly those which involved superficial outcome assessments or poorly generalisable samples (Chase, 2002; Jordans et al., 2011; Adhikari et al., 2018; Antle et al., 2018), should be interpreted with caution.

### Availability

The number of interventions conducted in different geographical regions assimilates the patterning of prevalence data, with most research activity in India, and substantially less work conducted in Afghanistan, Iran and Pakistan. Whilst access to formal mental health support is most limited in rural areas (Palit and Bandyopadhyay, 2016), the majority of studies in the current review were located in urban districts, indicating current community-based provision does not wholly address the lack of formal mental health infrastructure. Consequently, broad policy approaches may consider how pre-existing programmes can be applied across rural locations in South Asia.

Most studies were delivered in schools, with outpatient clinics, home-based programmes and community centres utilised to a lesser extent. School settings were noted as valuable for accessing large numbers of children at regular intervals to both test and implement intervention programmes (Berger and Gelkopf,

**Table 2.** Overview of bias risk using the Cochrane Risk of Bias Tool where applicable

| Author(s) | Quality assessment and bias risk[a] |
|---|---|
| Abdollahian et al. (2013) | Low risk<br>High quality |
| Adhikari et al. (2018) | Low risk<br>Medium quality |
| Andrew et al. (2020) | Low risk<br>High quality |
| Antle et al. (2018) | Low risk<br>Medium quality |
| Becker and Gelkopf (2007) | Not applicable |
| Berger and Gelkopf (2009) | Low risk<br>High quality |
| Blake et al. (2017) | Low risk<br>Medium quality |
| Borja Jr. et al. (2019) | Not applicable |
| Chase (2002) | Not applicable |
| Divan et al. (2019) | Low risk<br>High quality |
| Duggal et al. (2020) | High quality |
| Gonsalves et al. (2019) | High quality |
| Jordans et al. (2011) | Medium quality |
| Palit and Chatterjee (2006) | Medium quality |
| Perera et al. (2016) | Low risk<br>Medium quality |
| Puthanveedu and Sekar (2019) | Low risk<br>High quality |
| Rahman et al. (2016) | Low Risk<br>High quality |
| Rajaraman et al. (2012) | High quality |
| Zachariah et al. (2018) | Medium quality |

[a]For all applicable studies, a critical appraisal based on study strengths and limitations is given using the Cochrane Risk of Bias for RCTs and non-RCTs is also used.

2009). However, it is also necessary to acknowledge that some children in LAMIC South Asian settings are unable to attend school, often for reasons which also increase their mental health risk, such as poverty or an unstable home environment (Jordans et al., 2011). As such, it is important to consider the most appropriate avenues to address the needs of hard-to-reach groups.

A home-based programme for autism demonstrated equivalent efficacy to that conducted in specialist therapeutic settings, suggesting that the lack of formal care centres in rural settings can at least be partially addressed by upskilling local workers to deliver support (Perera et al., 2016). Using community centres was also noted to catalyse cohesion by bringing together different religious and subcultural factions (Chase, 2002). Grassroots community projects also hold particular value, because they are more sustainable, responsive and organic than top-down interventions which have a less flexible structure, and do not necessarily respond to the specific needs of the populations they are designed to support (Chase, 2002). This suggests that community-based interventions may be more effective in responding to situational factors (such as

increasing access to mental healthcare whilst reducing stigma) than larger-scale programmes.

## Treatment focus

Existing data suggest that the most common mental health disorders in South Asia are depression, stress, PTSD and anxiety (Hossain et al., 2020). In the present review, there were a high proportion of studies which investigated care for PTSD, an addition to a large body of work on autism. However, much less work attended to the treatment of depressive disorders. Ogbo et al. (2018) argue that the prevalence of depression in South Asia positions its prevention and treatment as a key priority, and as such, a greater degree of depression-related intervention may be a valuable direction for future research. Evidence also indicates that marginalised populations are most likely to present with mental health problems (Hossain et al., 2020). Five of the 19 reviewed studies concerned populations who typically suffer prejudice, including children of Devadasis and socioeconomically deprived communities. The Devadasi system is a practice followed by a small community in South India, which involves dedicating girls from low social castes to a deity and auctioning their virginity. Devadasi women are prohibited from marrying and their illegitimate children are shunned by society. On one hand, work that addresses the needs of those most likely to experience mental health problems is undoubtedly valuable. However, there are also data to suggest that distinguishing groups who are already stigmatised can exacerbate the effects of prejudice, and further isolate individuals from the rest of society (Puthanveedu and Sekar, 2019). Additional work suggests that community-based mental healthcare can have further unintended negative consequences, given by nature it does not involve specialist personnel and therefore must be carefully delivered. For example, in the present review, Berger and Gelkopf (2009) note that directly addressing trauma in children who had experienced exposure to terror and war resulted in increased sensitisation and the need for specialist attention. Equally, whilst Zachariah et al. (2018)) found that a suicide-prevention programme delivered by school children to their peers provided benefits for both providers and recipients, they also discovered peer educators could attempt to provide advice too quickly, causing recipients to feel misunderstood, and become withdrawn. Finally, after developing and evaluating a smartphone game to support adolescents at risk of depression in India, Gonsalves et al. (2019) noted that whilst the game was well received and effective, it could not completely replace direct counsellor supervision. These findings indicate the value of carefully targeting specific disorders and populations, in order to avoid exacerbating pre-existing problems.

In reducing the risk of negative intervention consequences, it may valuable to further investigate the approaches of strength building and proportionate universalism. In contrast to deficit-driven models, strength-based approaches aim to build resilience by developing human capacities which can protect against adversity (Gable and Haidt, 2005; Wong, 2013). Given that strength-based approaches shift focus away from formal diagnoses or treating disease symptomatology, they are less likely to risk stigmatising individuals who become labelled as 'ill'. Instead, strength-based approaches concentrate on the promotion of positive functioning across societies and individuals (Ryff and Singer, 2006). One study explored the use of creative healing, such as art- and drama-based therapy, as an alternative to medicalisation or counselling (Chase, 2002). At present, the evidence base for such work is limited, both in

South Asian and Western settings (Baker et al., 2018). However, given that non-medicalised methods hold potential for promoting well-being without evoking stigma, future research may consider investigating the efficacy of creative healing and other strength-based approaches in South Asia. Indeed, the growing emphasis on preventative healthcare methods in Western settings represents a learning opportunity at the global scale, suggesting South Asian countries may benefit from embedding preventative care at an early stage, as they develop their mental health resources.

The principle of proportionate universalism argues that interventions to reduce inequality are most effective when universally implemented, however with a scale and intensity that is proportionate to the level of disadvantage (Marmot and Bell, 2012). This model involves delivering projects that offer holistic treatment at scale, with targeted intervention for those in need of more specialised therapy. It increases the accessibility and cost efficiency of mental health support, whilst also acknowledging some groups require more extensive treatment, and has been applied to great effect regarding health inequalities in Western settings (Carey et al., 2015). A study conducted in Nepal (Adhikari et al., 2018) noted that a family- and school-based intervention to improve child behaviour problems could lead to successful results by providing children of lower social castes and additional services as a means of subtly addressing ingrained discrimination. Moreover, a study found that a multilayered programme which implemented low-intensity interventions at scale, as well as a triage system for those with more severe needs, efficiently matched the requirements of individual children to different levels of care (Jordans et al., 2011). This is pertinent given that resources to provide mental health support are particularly limited in South Asian settings, and therefore must be distributed wisely. As such, future research should continue to explore the integration of proportionate universalism amongst developing mental healthcare in South Asia.

### Feasibility

#### Task-shifting

Studies in the present review included a broad range of implementation methods, and the concept of task-shifting has been noted as particularly relevant for LAMIC settings, increasing both the practical and financial accessibility of projects by maximising available human resources. Task-shifting was utilised in 11 of the 19 identified articles, and a number of implications can be inferred across studies. For example, several authors perceived task-shifting as useful, including Rahman et al. (2016) and Rajaraman et al. (2012), who noted the value of this approach in scaling-up school health promotion in LAMIC settings. Moreover, Andrew et al. (2020) specifically described task-shifting as a factor in the success of a home-based intervention for child development in Indian urban slums, whereas Becker (2007) concluded that task-shifting is of distinct value in disaster settings because it allows large numbers of individuals to be trained and mobilised for prompt mental healthcare delivery. Finally, Palit and Chatterjee (2006) and Zachariah et al. (2018) also reported the benefits of task-shifting to extend to those who are trained to deliver care by providing a sense of purpose and employment.

However, certain conditions are also required for task-shifting approaches to work effectively. Primarily, the appropriate supervision of support staff for retaining implementation fidelity and protecting the mental health of delivery staff themselves are required. Jordans et al. (2011) reported significant distress among volunteer facilitators who implemented classroom-based

interventions for mental well-being compared to counsellors who received extensive training and renumeration for their time. Addressing logistical difficulties and providing mental health support could help to reduce stress across service providers, as well as adapting workloads Jordans et al., 2011). This finding is complemented by the observations of Andrew et al. (2020), who recommended physical health workers who are provided additional psychosocial training must be supported so that mental health intervention does not detract from their existing work. Rahman et al. (2002) and Rajaraman et al. (2012) concluded that comprehensive professional supervision was essential in training and supporting nonspecialist personnel in delivering mental healthcare, whilst Rajaraman et al. (2012) indicated that implementing clearly defined referral procedures helped lay school counsellors to feel confident in delivering appropriate care, and to signpost serious cases when necessary. The importance of a hands-on yet flexible supervisory approach was also effective in upskilling professionals supporting children with developmental difficulties (Duggal et al., 2020). One study noted that facilitators on a peer-education programme to reduce suicide should be reminded of the limits of their personal responsibility, and provided opportunities for group debriefing and social bonding in order to prevent burnout (Zachariah et al., 2018). Lastly, in a study evaluating large-scale psychosocial support interventions for Rohingya refugees, the authors stated that projects that involve international experts must strive for stability in management, rather than short-term tenures, which can lead to changes in strategy and difficulty in the implementation of programmes (Borja Jr. et al., 2019). This can subsequently result in unfavourable responses from other sectors, who devalue inconsistent programmes as disorganised.

### Training

The importance of providing training for mental health provision in South Asian settings is emphasised to be important for community-based interventions. One study (Antle et al., 2018) suggested that training parents to deliver interventions is unrealistic in some settings, due to deficits in literacy, time and educational background. Alternatively, a home-based early intervention programme for autism conducted in urban Sri Lanka (Perera et al., 2016) found that offering parents' individual supervision, demonstrating activities and materials, conducting follow-up sessions and providing personalised information strongly contributed to the programme's success, via increased motivation and capacity of parent interventionists. Becker (2006) noted that the adoption of a train-the-trainer model is valuable in resource poor settings, specifically in the training of large numbers of people. Under this model, initial training is delivered to a small number of individuals who then go on to impact organisations. Given the importance of best understanding how to recruit, train and supervise those who deliver task-shifting approaches (Borja Jr. et al., 2019), future research may consider these factors in more detail, as a means of bridging the treatment gap between need and availability of care LAMICs.

### Telepsychiatry

It is also valuable to consider the growing use of technological approaches, particularly given the transformative effect of COVID-19 on telemedicine (Portnoy et al., 2020). Telepsychiatry is an emerging resource in South Asian settings, particularly in India where the rapidly increasing penetration of mobile devices enables geographically unrestricted intervention. Smartphone games can be useful in promoting engagement, decreasing stigma

and addressing some of the challenges of delivering accessible and timely mental health support in under-resourced society. However, Gonsalves et al. (2019) concluded that such tools should be extensively tested to ensure efficacy and acceptability, and paired with instructional and relationship support from physical counsellors. Two studies also explored how technology can be used to successfully address difficulties in training for mental healthcare. Antle et al. (2018) investigated the use of a computer-tablet game to improve self-regulatory skills, and found that participants quickly and successfully learnt how to use the platform, removing the need for parent training on the delivery of mental health support. The efficacy of the game was observed in real-world change, including decreases in hyperactive, aggressive and anxious behaviour, and increases in self-esteem, concentration, self-expression and discipline. Duggal et al. (2020) explored the use of technology in training a broad range of child health practitioners to develop autism-specific treatment skills. They found that online supervision enabled continuity and interaction across participants, and facilitated the involvement of multidisciplinary individuals from a number of geographical areas. This is highly significant in LAMIC settings, where professional awareness of autism is low, training opportunities are limited and experts are localised to a few locations. However, it is also important to note that access to technology may still be limited in some areas or populations, and to integrate such contextual information in the design or adaptation of interventions.

### Acceptability

#### Adaptation of materials

One of the most prevalent observations was the importance of adapting intervention materials to specific contextual requirements, which have been highlighted by several authors. There is a lack of pre-validated survey instruments that have been reliably translated for South Asian contexts; many tools contain questions inappropriate for children living in poverty (Antle et al., 2018). Altering designs in smartphones to account for literacy difficulties across youths in India was found to significantly improve literacy, and by designing the game to operate offline allowed youths to engage with the game in regions with limited Internet access (Gonsalves et al., 2019). Another example is demonstrated by a study conducted in Sri Lanka. As emotions are widely expressed via body processes, and acknowledging one's strength is public is considered ostentatious, the authors adapted their programme to emphasise somatic experience and encourage children to take pride in their strengths whilst acknowledging weaknesses (Berger and Gelkopf, 2009). The importance of attending to the meaning and relevance of terms across languages to ensure comprehension was also recommended (Borja Jr.. et al., 2019), and to use qualitative methods to understand the context in the development and implemented on materials (Blake et al., 2017). This is particularly important during projects that involve the consecutive translation of materials across multiple languages, and in settings where concepts related to mental health are often new and not widely understood. Future research may consider how intervention materials can be best developed to account for cultural factors, as well as the development of questionnaires which are appropriate for research with target populations.

#### Relationships across stakeholders and researchers

Intervention acceptability was also influenced by the relationship between researchers and stakeholders. In a school-based

programme which trained community volunteers as lay counsellors, teachers initially mistakenly referred children to the programme based on disciplinary rather than mental health problems, and complained about a lack of information sharing (Rajaraman et al., 2012). These issues were substantially reduced after monthly updates were produced and disseminated amongst the faculty, resulting in a more positive evaluation of the programme across both teachers and students. The reports were also effective in reducing the stigmatising attitude of some teachers, and identifying a specific period to deliver counselling in the school timetable further alleviated tension between counsellors and staff. When delivering mental health training to those already working in the field, Duggal et al. (2020) stated that incorporating stakeholder feedback is particularly critical in South Asia, because professionals have often received some basic training, but lack specialised skills, which require deliberate practice and reflection.

### Intervention terminology

Lastly, several authors noted that participants appreciated interventions which did not require medical professionals, or the use of diagnostic terms. In particular, the substitution of the word 'ASD' with 'problem behaviours' promoted engagement by parents of children with autism by emphasising an outcome parents were most motivated to address, and reduced the fear of possible stigma related to labelling a disability (Blake et al., 2017). Likewise, another study investigated a programme to provide cognitive-behavioural play therapy for children with ADHD, and parents welcomed a form of therapy which did not involve referral to a psychologist, psychiatrist or physician for medical treatment (Abdollahian et al., 2013). Finally, Puthanveedu and Sekar (2019) reported that children of devadasis are stigmatised to the extent they are isolated to outlying districts where facilities are poorly available. As such, providing community-based mental health provision in these regions helps to avoid the entrenchment of further discrimination, and may support the eventual 'mainstreaming' of marginalised populations.

## Discussion

The present search revealed only a small number of studies regarding community-based intervention for children and adolescents living in South Asia with most research activity in India. This is unsurprising, given known deficits in research and practice for mental health support amongst this population (Kieling et al., 2011). However, the results do indicate the emerging existence of community-based youth mental health provision in South Asia, including several studies which are the first of their kind to be undertaken in a LAMIC setting. For example, Rahman et al. (2002) presented a novel contextual evaluation of an ASD intervention, and Jordans et al. (2010) investigated a pioneering multilayered psychosocial healthcare initiative. We found that the majority of the included studies were (1) conducted in India, (2) delivered in school-based and urban settings and (3) focused on the treatment of PTSD or ASD.

The empirical quality of identified studies was variable, with a high prevalence of small and poorly generalisable samples, and low use of adequate comparators or pre-validated assessment materials. These factors likely result from the complexity of conducting research in LAMIC South Asian settings; resources are generally constrained, and the availability of questionnaire materials to assess mental health outcomes which are comprehensible to populations

of interest is low. Moreover, there can be logistical and administrative complications regarding the implementation of randomised controlled trials in South Asian settings, as well as the ethical dilemma of withholding treatment to control groups (Becker, 2006). Therefore, future research could also address the development of contextually appropriate questionnaires, and enhance pre-existing research by utilising larger and more diverse samples.

It is also important to acknowledge cultural and linguistic factors to support accepted and effective intervention. Whilst Fricchione et al. (2012) call for an increase in research at the local context level to adapt and evaluate interventions to ensure that it is appropriate for the context, it is important to understand the scale of language barriers. South Asia is one of the most diverse regions in the world, and each of the countries in South Asia is rife with cultural and linguistic diversity. India, for example, has 22 major languages (Department of Higher Education, 2021). Interventions and translation of materials must be innovative to cope with the complexities of the diversity of language and culture (Patel and Prince, 2010).

Another issue with linguistic barriers arises from the communication difficulties between Western professionals and health workers, where interpreters may be used. Swartz et al. (2014) argue that using interpreters to translate across cultures may not be sufficient to bridge this gap due to the vast differences in languages and cultural nuances, and translating instruments into a few local languages are insufficient to meet the diversity and complexity of language. It is also important to recognise the vast differences in urban–rural and socioeconomic settings across the regions, where materials may need to be adapted. Literacy levels within countries also differ considerably adding on to the complexity of translation of intervention materials which needs to be adapted to be suitable to the target users.

There should be ongoing dialogue and discussions between Western professionals/clinicians, researchers and lay workers on the ground who are familiar with the cultural and linguistic context of the families and target users during the adaptation of intervention materials. For example, Blake et al. (2017) report training received by the native clinicians in ASD by Western professionals, and ongoing discussions between investigators from Bangladesh and the United States, and native clinicians in Bangladesh to ensure that the materials used for an ASD intervention were suitable for the cultural, linguistic and socioeconomic context. Language was simplified, and where possible images were used rather than words to accommodate parents with low literacy levels. Images were also appropriate within the specific rural and socioeconomic context. Importantly, there was ongoing dialogue between the native clinician delivering the intervention, study investigators and scientists involved to ensure that the material was culturally, linguistically and contextually appropriate while ensuring that the concepts are not lost in translation. Swartz et al. (2014) also call for an increase in reports and sharing knowledge by workers who have worked on the ground and developed and delivered mental health programmes, overcoming language barriers.

Common to many LAMIC regions, it is also important to consider the power imbalance between Western professionals and interpreters and lay workers during training, where they may feel compelled to behave in a way that they think is most acceptable to professionals, thus compromising the efficacy of delivery (Swartz et al., 2014). Power imbalances are also observed between professionals or health workers and participants in a study where participants may not fully answer questions due to what they perceived was a lack of education on their side

compared to the professional, and therefore needed more probing to provide their perspectives (Roberts et al., 2020). It is important to be aware of these cultural differences when communicating and interacting with participants who may feel compelled to behave in a way that they perceive is appropriate to the professional.

In summary, the paucity of resources and trained professionals in South Asian settings leads to a significant gap between mental health treatment and needs. Capacity to address this via professional training is limited, given the resource-constrained context of South Asia, and so utilising methods which are innovative, flexible and make best possible use of existing capacity is critical. Community-based mental health provision appears to provide a potential answer to this need, particularly when task-shifting and stigma reduction methods are implemented. However, there needs to be ongoing support in relation to supervision of staff to maintain effectiveness of intervention in the long term. This support should incorporate training, mentoring and capacity development in order to ensure a more sustainable model for intervention implementation.

The results of this review indicate the availability of mental health support, and can also be used to understand the feasibility, acceptability, impact and quality of existing interventions. Additionally, these findings provide new insight on approaches which are thought valuable for South Asian settings, primarily task-shifting and stigma reduction, with implications for policy, practice and research. This review also provides a number of directions for future research to build upon existing intervention, in order to bolster the mental health and future prosperity of South Asian societies.

## Limitations

It is important to note several limitations of the current work. First, we must acknowledge that mental health is a complex concept, requiring interdisciplinary intervention. Our approach is focused upon the infrastructure and enactment of mental health treatment, rather than 'care' in its most comprehensive term, and therefore should be viewed as complementary to additional investigations which utilise different disciplinary perspectives, for example, those that consider community-based interventions integrated amongst care programmes.

Additionally, our study is limited only to the articles in the review which was determined by the inclusion and exclusion criteria. For example, we exclusively considered studies where the mean age of samples is between 0 and 18. Studies that may have included slightly older young adults may provide information on interventions that are applicable to those under 18; however, this work was not included in the current review.

The review was also dominated by studies in India, and given the cultural and geographical diversity of South Asia, our findings are limited in their extrapolation to every region considered. However, this is reflective of relatively developed mental health infrastructure and research capacity in India compared to other countries, and therefore provides important detail on the current state of research output in other regions. Relatedly, we sought to obtain insights from grey literature including reports and dissertations that may include important insights on community-based care in this region. Despite contact with local agents, any material proved highly difficult to access, and therefore could not be included in the present review.

## Recommendations for practice and research

The present review indicates that community-based mental health provision for children and adolescents in South Asia is meagre, yet emerging. Identified studies suggest that community-based interventions can be effective, acceptable and impactful for young South Asian populations, and may hold particular value in reducing the impact of stigma and its consequences, whilst addressing the paucity of care available outside of urban districts with formal centres of care. The findings presented here suggest that task-shifting approaches should incorporate comprehensive and flexible supervision, clearly defined referral procedures and a consistent managerial team. Results also indicate that community-based programmes may be most effective when complemented with targeted specialist intervention, as recommended in proportionate universalism. Furthermore, the contextual adaptation of intervention materials is critical, and must take into account linguistic, cultural and logistical factors. One approach taken in 'implementation science' methodology is to co-design and co-produce interventions with the end user, and this seems particularly appropriate here given the diversity in intervention context (Eccles et al., 2009).

Priority areas for further investigation are to develop research capacity and practice in rural locations and in countries with a relatively reduced output, as well as to increase the number of studies which investigate the treatment of depressive disorders. Work on the efficacy of community-based intervention in reducing stigma could also be complemented by considering strength-based approaches. Whilst task-shifting methods appear to present strong value for mental health intervention in low resource settings, it is important to acknowledge that ongoing support is critical, and greater knowledge regarding the supervision, training and recruitment of staff across different contexts and populations could further support their implementation. Additionally, consideration should be given to the sustainability of support after initial research projects or external funding has ended. Lastly, technology could be utilised to a greater extent to address the treatment gap between demand and provision of mental health support, and as such further research should build upon the investigations included here.

**Open peer review.** To view the open peer review materials for this article, please visit https://doi.org/10.1017/gmh.2022.49.

**Data availability statement.** Data availability is not applicable to this article as no new data were created or analysed in this study.

**Acknowledgements.** We would like to thank the Mental Health Initiative in South Asia (MhiSA) partners, the facilitators and delegates from the 'South Asia Collaboration on Mental Health' event held in Kathmandu, Nepal, on 10–12 April 2018 in which the discussions held during the event led to this scoping review: Dr Fatemeh Ahmadi (British Council, UK), Dr Javad Fadardi (Ferdowsi University of Mashhad, Iran), Dr Rebecca Crane (Bangor University, UK), Dr Farzana Islam (Child Development Centre, Apollo Hospital, Bangladesh), Dr Sanjeev Jain (National Institute of Mental Health and Neurosciences, India), Dr Sumeet Jain (The University of Edinburgh, UK), Mr Suraj Koirala (Transcultural Psychosocial Organization, Nepal), Dr Anand Krishnan (All India Institute of Medical Sciences, India), Mr Nagendra P. Luitel (Transcultural Psychosocial Organization, Nepal), Dr Kedar Marahatta (World Health Organization Country Office, Nepal), Dr Mohammad Nami (Shiraz University of Medical Sciences, Iran), Dr Jacqueline Rodgers (Newcastle University, UK), Dr Rajesh Sagar (All India Institute of Medical Sciences, India), Dr Sayyed Ali Samadi (Ulster University, UK), Dr Aditya Sharma (Newcastle University, UK), Dr Rahul Ramchandra Shidhaye (Public Health Foundation of India), Dr Michaela Swales (Bangor University, UK), Dr Zahra Tabibi (Ferdowsi

University of Mashhad, Iran) and Dr Fahmida Tofail (International Centre for Diarrhoeal Disease Research, Bangladesh).

**Author contributions.** Conceptualisation: R.A.S. and J.A.P.; Writing – original draft: R.A.W.; Writing – review and editing: all authors.

**Financial support.** This scoping review was supported by the British Council.

**Competing interests.** The authors declare no competing interests exist.

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
