## [Reviewer Report]

Dr Azlina Amir Kassim

Lecturer

School of Human and Behavioural Sciences

Bangor University

July 22, 2022 

Professor Gary Belkin (Editor-in-Chief) and 

Dr Rahul Shidaye (Associate Editor)

Cambridge Prisms: Global Mental Health

Dear Prof Belkin and Dr Shidaye,

Thank you for the opportunity to submit a revised draft of the manuscript. We thank you and the reviewers for their time and feedback on our manuscript. We have worked hard to address the reviewers’ comments in what we believe is the final version of the paper. We hope the manuscript will now be acceptable for publication in Cambridge Prisms: Global Mental Health. In addition to the revised paper, we also include the graphical abstract, impact statement and social media summary with this submission. 

Below, we detail how we have responded to the reviewer’s comments (in bold). All edits in the main manuscript in line with the reviewers’ comments are in red font. We also include the revised sentences and edits in this letter. 

• Reviewers comment 1: Page 4: "in India, Bangladesh, Pakistan or Afghanistan (Belfer, 2008), and very few in Nepal (WHO, 2017)." These new precisions are appreciated but one of the citations is 14 years old and the other is 5 years old. Especially for the Belfer citation (the older), one wonders if the information remains accurate?

<b>We have revised the sentence and updated the citations to reflect the more current situation, as below:</b>

Revision (page 3): “In Bangladesh, Sri Lanka, Nepal and Pakistan there are no child-specific mental health policies (WHO, 2017, Hossain et al., 2021). Additionally, very few psychiatrists offering specialised treatment for children or adolescents in India, Bangladesh, Pakistan, Nepal or Afghanistan (Agarwal, 2021; Hamdani, Huma, & Tamizuddin-Nizami, 2021; Singh, Gupta, Singh, Basnet, & Arafat, 2022; Skuse, 2021; Wang et al., 2022).”

• Reviewers comment 2: Page 4: a poor psychosocial environment is listed as a risk factor - it would be good to understand what is meant by this. The list in general has a lot of overlapping concepts.

<b>We agree that the list contains overlapping concepts, and we have replaced the term “poor psychosocial environment” to “low educational stimulation” to be more specific. We believe that this paragraph is clearer now with this addition, and other psychosocial factors more clearly delineated in the subsequent paragraph. Revised text is below:</b>

Revision (page 4): “Established risk factors for the development of emotional, intellectual and behavioural disorders in LAMIC South Asian settings include low educational stimulation (Arun & Chavan, 2009),…….”

• Reviewer comment 3: There remains on pages 4-5 ambiguity about whether the lists of factors contributing to mental health risk or resilience are referring specifically to factors documented in South Asian countries or whether they are factors observed in other settings and hypothesized to be active in South Asia as well.

<b>We have added a sentence on page 5 to specify that these factors are observed across the globe and referred to evidence to their specific relevance in south Asia. The revised sentence is below:</b>

Revision (page 5): “These factors and their relationships to mental health risk are observed in numerous settings across the globe, and the aforementioned articles evidence their specific relevance to the development of mental health conditions in South Asia”

• Page 6: "This is particularly pertinent given specific observations that diagnostic and therapeutic services in primary care are lacking in India and Pakistan (Karim et al., 2004; Khandelwal et al., 2004; Regmi et al., 2004)". Again these citations are 18 years old; it would seem important when making statements about the presence or absence of particular services to have some more recent data or at least an assertion that there is no reason to believe that the situation has changed.

<b>We have updated the statements with recent citations. This updated statement is below:

</b>

Revision (page 6): “This is particularly pertinent given specific observations that diagnostic and therapeutic services in primary care are lacking in India and Pakistan (Agarwal, 2021; Hamdani et al., 2021)…..”

We thank you again for your time and feedback, and for consideration of this revised manuscript for publication. For any further information, please do not hesitate to contact the undersigned author. 

Yours sincerely,

Azlina Amir Kassim

a.amirkassim@bangor.ac.uk

---

## [Reviewer Report]

*Comments to Author*: Round 1 

Recommendation - Major revisions

The authors of this review have tackled an important area (child mental health services) in an important part of the world. The review makes some important points about those services and the opportunities and barriers to providing them. I think that the manuscript could be strengthened with attention to a few points:

Page 4: not clear (end of first full paragraph) whether the text refers to the region or more generally; same on page 5 though eventually clear that is more general. It could be helpful to clearly start with a more general pre-amble and then talk about what is known to date about service needs in South Asia.

Page 5: I think that there are some definitional issues here that a) may not reflect how terms are used by other investigators and b) may lead to some confusion about the search parameters. Are all community-based programs provided by non-specialists necessarily (this is repeated on page 26)? In this review, 12 of the 17 studies found involve mental health professionals (page 10); one of the included studies from Iran, for example, is set in an outpatient mental health clinic. Does the introductory section need to be refined to widen the focus to all programs that could be thought of as “in the community” whether they involve task shifting to generalists or peers or not?

Page 5; Does all task-shifting involve community-based programs? Tasks can be shifted in many settings. Does all task shifting have to be limited to psychosocial services versus prescribing? In adult collaborative care, many task-shifing programs focus specifically on prescribing.

Page 6: Do the investigators have any hypotheses about why community-based mental health programs in South Asia might face challenges that are similar to or different from programs in other regions? In addition, the range of countries is large – with widely varying health systems and contexts that involve both low and middle income (and even some high income) regions of those countries as well as populations that are in the middle of various forms of humanitarian crisis (sanctions, internally and externally-displaced populations). What themes might possibly be similar or different across these contexts?

Page 7: In the discussion or methods, it could be important to talk about the possible impact of some of the review exclusions; one could imagine some programs for older teens that could possibly focus only prescribing; there could be collaborative care programs involving GPs or equivalents that might involve both youth and adults and have a mean age >18 but reasonable subgroup analysis of younger patients.

Figure: On page 6 say that “all study designs’ are eligible but in the figure, 94 studies are said to be excluded because of design. This might bear a brief explanation

Page 10: The authors state: Task-shifting methods were also employed in 11 articles, whilst three studies used a train-the-trainer approach, and two papers investigated parent-mediated interventions. This is potentially confusing, since TOT (it would depend on who is being trained) or parent-mediated interventions could be part of task-shifting. That is, TOT and training parents can be mechanisms used to accomplish task shifting (thinking that task-shifting has a target provider, a task, and then a way of having a specialist or expert train and support the target provider in the task).

Page 10 under design: I am not sure how to interpret this sentence: "The most commonly used experimental design was a randomised controlled trial (RCT) design (n=6), followed by a controlled before-after (CBA) study (n=4). In two articles, this was supported with descriptive qualitative data, in addition to eight studies which employed a descriptive design only and one case study. Most experimental articles did not contrast treatment groups with any kind of comparator (n=11)." If there were 19 studies, of which 10 were either RCT’s or controlled before-after, how could 11 have no comparator? Do the authors mean “no other defined form of treatment” (versus wait list or treatment as usual)?

Discussion: much of what is in the discussion would seem to belong in the results section. The results describe the studies, but what a reader would like to know is what lessons seem to have been learned, especially about the opportunities and barriers to developing community-based child mental health interventions in general and in particular in this region. Two universally applicable messages seem to be a) that individuals who take on task-shifted interventions need ongoing support, and that b) programs need to set up mechanisms to provide this support once the “outside” experts have gone. The authors note that there are also barriers related to literacy and competing demands among the targets of task-shifting. It would be interesting if the authors could comment about whether there are aspects of these barriers that are different in South Asia compared to other regions.

The authors speak relatively little about the cultural or linguistic context in which the studies they identified are taking place. As an example, on page 26, the authors describe some important potential negative impacts of interventions that could possibly be related to a low background level of mental health “literacy” (at least in Western terms; the authors note this very briefly on page 29) or high levels of stigma. How might the cultures of South Asia (or at least the ones represented in the review) influence the design and outcome of community-based services? Who might be the likely targets of task shifting in this region – and how might that be different than in other countries?

Date submitted: 5-Feb-2022

Round 2

Recommendation - minor revisions

Thanks to the authors for their thoughtful responses to the reviewer comments. Some thoughts on this version:

Page 4: "in India, Bangladesh, Pakistan or Afghanistan (Belfer, 2008), and very few in Nepal (WHO, 2017)." These new precisions are appreciated but one of the citations is 14 years old and the other is 5 years old. Especially for the Belfer citation (the older), one wonders if the information remains accurate?

Page 4: a poor psychosocial environment is listed as a risk factor - it would be good to understand what is meant by this. The list in general has a lot of overlapping concepts.

There remains on pages 4-5 ambiguity about whether the lists of factors contributing to mental health risk or resilience are referring specifically to factors documented in South Asian countries or whether they are factors observed in other settings and hypothesized to be active in South Asia as well.

Page 6: "This is particularly pertinent given specific observations that diagnostic and therapeutic services in primary care are lacking in India and Pakistan (Karim et al., 2004; Khandelwal et al., 2004; Regmi et al., 2004)". Again these citations are 18 years old; it would seem important when making statements about the presence or absence of particular services to have some more recent data or at least an assertion that there is no reason to believe that the situation has changed.

Date submitted: 28-Apr-2022

---

## [Reviewer Report]

*Comments to Author*: Round 1

Recommendation - minor revisions

This is an important paper and I enjoyed reading it. You make important contributions in synthesizing evidence in this underdeveloped area. The policy and research implications are clearly articulated.

A few comments:

- I did not see a clear rationale for why a scoping review. Would be helpful to also discuss this methodology, possibly with reference to guidelines like those from the Joanna Briggs Institute. Linked to this, it is important to clearly state the aims of the review.

- It is crucial to have some discussion on authorship in relation to current GH debates on decolonziation and authorship would be important given the absence of any author from a South Asian university. This could link to a section on personal and professional positionalities of the authors and project and how this will have shaped the study.

- In the methods would be useful to discuss the relevance of the search terms. What literature might have been excluded by the use of largely biomedical and psychological terminologies?

- Search terms – would be important to provide more details of these and the list of terms used. This would help the reader to also understand the limitations of the study.

- A limitations section is crucial: we don’t really get to know the broader and specific limitations of the study. For example, the study doesn’t appear to target studies addressing social and community development interventions that might be important in improving mental health of young people. It would be useful to understand these limitations; but also to understand decisions taken and the implications of these. Similarly, the grey literature is not really considered; and the reasons for this should be discussed (i.e. difficulties in access).

- Conceptually, it would be useful in the limitations to discuss how disciplinary perspectives shaped the approaches. For example, the approach is clearly focused on 'treatment systems' in the community rather than care systems. These underlying assumptions should be surfaced either in the limitations section or another aspect of the methods section. This would help make the paper more relevant to an interdisciplinary GMH audience.

Date submitted 13-Feb-2022

---

## [Reviewer Report]

*Comments to Author*: Round 1

One of the reviewers has requested to clarify search definitions and separate conclusions of the review from the discussion. The other reviewer has expressed concern about the lack of any author from South Asia. Kindly address these important concerns along with other comments from the reviewers. 

Date submitted: 14-Feb-2022

Round 2

Date submitted: 29-Jun-2022